# Current Utility of Arbuscular Mycorrhizal Fungi and Hydroxyapatite Nanoparticles in Suppression of Tomato Root-Knot Nematode

Saad Alamri [1], Nivien A. Nafady [2], Atef M. El-Sagheer [3], Mohamed Abd El-Aal [4], Yasser S. Mostafa [1], Mohamed Hashem [1,2,*] and Elhagag A. Hassan [2,*]

[1] Biology Department, Faculty of Science, King Khalid University, Abha 61321, Saudi Arabia; saralomari@kku.edu.sa (S.A.); ysmosutafa@kku.edu.sa (Y.S.M.)
[2] Botany and Microbiology Department, Faculty of Science, Assiut University, Assiut 71516, Egypt; niviennafady@aun.edu.eg
[3] Agricultural Zoology and Nematology Department, Faculty of Agriculture, Al-Azhar University, Assiut 71526, Egypt; atefelsagheer@azhar.edu.eg
[4] Chemistry Department, Faculty of Science, Assiut University, Assiut 71516, Egypt; mohamedabdelaal@aun.edu.eg
* Correspondence: mhashem@kku.edu.sa (M.H.); elhagaghassan@aun.edu.eg (E.A.H.)

**Abstract:** Effective biosafe management strategies are used to decrease world crop damage produced by plant-parasitic nematodes. This study evaluated the efficiency of hydroxyapatite nanoparticles (n-HAP) and mycorrhizal fungi to control the *Meloidogyne incognita* infecting tomato plants. Application of n-HAP significantly increased the juveniles' mortality (195.67%) and egg hatching inhibition percentage (80.71%) compared to the untreated control, in vitro. Mycorrhizal and/or n-HAP treatments increased the plant growth parameters (root and shoot length, dry weight, and leaf area) and reduced the negative consequence of nematode infection. This may be due to indirect mechanisms through increasing plant nutrient uptake efficiency and increasing the internal plant resistance against nematode infection. In dual-treated plants, phosphorus, nitrogen, and calcium content recorded the highest value in the nematode-infected plants. Whereas the dual inoculation significantly increased mineral contents in tomato plants compared with control, this may induce the strength of the cell wall of the epidermal layer and cortex and consequently increase the plant resistance against nematode infection. Our results revealed that the application of the plant resistance-stimulants enhanced the plant growth parameters and internal nutrient content and reduced the nematode's criteria. Consequently, the internal plant resistance against nematode infection was induced.

**Keywords:** mycorrhizal fungi; hydroxyapatite nanoparticles; *Meloidogyne incognita*; nematode infection; plant resistance-stimulants

## 1. Introduction

Plant-parasitic nematodes (PPNs) or phytonematodes are one of the multiple causes of soil-related sub-optimal crop performance [1]. Through the soil-inhabiting organisms, the plant-parasitic nematodes dominate over pathogens after fungi in causing a permanent damage to the host, control, and sustainability of management [2]. Globally, crop loss induced by PPNs have been evaluated at USD 80 billion annually in open cultivations [3]; however, in protected cultivations, it might exceed 30–60% [4]. Root-knot nematodes (RKN) as sedentary endoparasites of the genus *Meloidogyne* (Tylenchida: Meloidogynidae) is considered one of the most important genera of nematodes that cause agricultural damage [5]. Considering the increasing global need for food production through intensive agriculture, traditionally, the agricultural damages caused by RKN have been managed by chemical practices. This causes defect to the environment and public health in the long run, in addition to the high economic cost of nematicides [6]. Therefore, it was necessary to search

for alternative methods to combat RKN by looking at the interaction of RKN with the biological content of the soil [7]. Microbial antagonists as biological control agents of nematodes are one of the potential alternatives to chemical nematicides; however, they are dependable under various environmental conditions. Therefore, attention should be paid to its relationship to plant in an unconventional form, to find or create an antagonistic relationship between RKN and some soil microorganisms to induce plants against nematodes [8]. One of the most promising relationships in this regard is using mycorrhizal fungi and nanotechnology-based formulations [9].

Arbuscular mycorrhizae (AM) are fungi of the phylum Glomeromycota [10] that form a symbiotic relationship with the roots of the host plant providing nutrients and can protect the host plant against abiotic and biotic stress factors. Native mycorrhizal fungi strains are used as biofertilizers and bioprotective agents to increase plant's yield and protect the plant from the pathogens, as an ecofriendly agent; however, the inoculation of plants with many mycorrhizal species belonging to different families were found to be more efficient than mono-inoculation. The co-inoculation showed a good impact on the mineral nutrition and the tolerance of plants against both biotic and abiotic stresses [11]. Plant-parasitic nematodes and AM fungi colonized host plant roots as a source of host photosynthate, a site for their existence and reproduction [12]. The interactions between mycorrhizal fungi and nematodes are complicated and depend on the soil conditions, fungal species, host plants, and nematodes [13]. Mycorrhizal-nematode interaction is prospected to increase the resistance of plants to nematode infection. Several authors reported that mycorrhizal fungi could activate plant systemic resistance to resist possible nematode invasion [14–17]. Alternatively, mycorrhizal fungi may have a direct mechanism to suppress nematode root infection by competing for feeding and root space.

Nutrient use efficiency (NUE) is based on the ability of plants to absorb nutrients from the soil, also on the storage and remobilization of nutrients [18]. NUE is a very important factor for evaluating plant yield. Due to intensive agriculture, change in physical and chemical properties of the soil, and change in the type of fertilizers used, the NUE of macronutrients is less than 50% [19]. High crop yield with good quality, soil health, and water or sustainable environment are the main determinants of food security, which has enabled scientists to discover new advances in nanotechnology [20]. Nowadays, the application of phyto-nanotechnology in agriculture is considered an ecofriendly method which has the prospective tools to improve crop production. Also, the development of nanoparticles for agriculture should be complemented by biosafety and eco-toxicological studies on the interactions between nanoparticles and the biological components of the soil [21]. Hydroxyapatite ($Ca_{10}(PO)_6(OH)_2$, HAP) nanoparticles have a promising possibility to be used as nano-fertilizer, which is used as a carrier of phosphorus fertilizer. HAP nanoparticles are the main inorganic component in the hard human and animal tissue. HAP nanoparticles have attracted a great deal of attention in several applications such as antibacterial agent [22], heavy metal, and dye removal [23,24] due to its excellent biocompatibility, bioactivity, high structural stability, high ion-exchange, and adsorption capacity. Therefore, when considering the application of nanoparticles to agriculture, the n-HAP should be ecofriendly and does not cause any negative impact on the environment and human health [19]. In this regard, the present study aimed to synthesize n-HAP, assessing for the first time their nematocidal effect on *Meloidogyne incognita*. Moreover, this study examined the effect of dual inoculation with nano-fertilizers (n-HAP) and native mycorrhizal fungi on tomato plants infected with *Meloidogyne incognita* under greenhouse conditions.

## 2. Materials and Methods

### 2.1. Synthesis and Characterization of Hydroxyapatite Nanoparticles (n-HAP)

HAP nanoparticles (n-HAP) were prepared by a precipitation method; 12.21 g of Ca $(NO_3)_2$.4$H_2O$ was dissolved in 500 mL of bi-distilled water, and the solution was stirred and heated to 70 °C for 30 min. A solution of $NH_4HPO_4$ (3.72 g in 200 mL of bi-distilled water) was added dropwise with vigorous stirring. The molar ratio of Ca/P was maintained at

1.67. The solution's pH was adjusted at 10 using ammonium hydroxide. After that, the formed precipitate was aged at 70 °C for 2 h. Next, the resulting HAP precipitate was filtered and rinsed three times with bi-distilled water and twice with ethanol, followed by drying at 80 °C for 24 h. Then, the HAP powder was calcined at 500 °C for 3 h in a static air atmosphere.

Characterization of n-HAP was performed by X-ray diffraction (XRD), Fourier Transform infrared spectroscopy (FTIR), and transmission electron microscope (TEM). XRD was performed using a Philips PW 2103 diffractometer (Netherland, CuK$\alpha$ = 1.54056 Å radiation source). Diffraction patterns were recorded over a 2θ range starting at 4° and finishing at 80°. The incremental step size used over the 2θ range was 0.06°. FTIR was carried out with a Nicolet spectrophotometer (model 6700) spectrometer (wavenumber range of 400–4000 cm$^{-1}$) using the KBr technique. TEM was carried out using the JEOL TEM (Model 100 CX II; Tokyo, Japan) at the Electron Microscopy Unit, Assiut University, Egypt.

### 2.2. Inoculum Propagation of Meloidogyne incognita (Kofoid and White) Chitwood

Naturally infected tomato (*Lycopersicon esculentum* L. cv. Pritchard) plant roots were collected from Assiut city, Egypt, and Abha city, Saudi Arabia. A pure stock culture of the root-knot nematode (RKN) was prepared from infected roots. Individual egg masses with their mature females were removed from root tissue. Each egg mass linked with the female was placed in a small glass capsule containing fresh water. Each female was collected and preserved in glass capsules containing 4% formaldehyde solution. Each mature female was identified to species by perineal pattern according to morphological features described by Chitwood [25] and Sasser [26]. The egg mass formed by *M. incognita* was hatched and re-inoculated to the seedlings of tomato plants according to the protocol described by Southey [27]. The infected roots were used as a source of inoculation for further experiments.

### 2.3. Inoculum of Mycorrhizal Fungi

The mycorrhizal inoculum used in the greenhouse experiment containing spores of *Rhizophagus aggregatus* [(N.C. Schenck & G.S. Sm.) C. Walker], *Funneliformis mosseae* [(T.H. Nicolson & Gerd.) C. Walker & A. Schüßler] and *Gigaspora gigantea* [(T.H. Nicolson & Gerd.) Gerd. & Trappe] was utilized after mass production in pots containing a sterilized soil and cultivated with maize as trap culture. The native mycorrhizal spores were recovered from the rhizospheres of plants by the wet sieve method [28]. The bio-inoculum (100 g soil) contained colonized root segments, extraradical hyphae, and 960–980 mycorrhizal spores.

### 2.4. Raising of Tomato Plants

In the greenhouse pot experiment, tomato seeds (*Solanum lycopersicum* L.), cultivar Hana, were used. Seeds were surface-sterilized for 2 min with 70% ethanol. Tomato seeds were washed several times with sterilized water and then planted in plastic cell (4 × 4 cm) plug trays filled with vermiculite under greenhouse conditions for 2 weeks.

### 2.5. In Vitro Assessment of HAP Nanoparticles' Effect on M. incognita Eggs Hatching and Juveniles' Mortality

About 200 eggs and 200 s-stage juveniles (J2s) were transferred to 5 mL of three concentrations of HAP nanoparticles (50, 150, 200 ppm), as separate treatments, in Petri dishes and covered under laboratory conditions (26 ± 2 °C), while distilled water served as a control, and each treatment was replicated three times [29]. The hatched second stage and dead juveniles (J2s) were observed at 24, 48, and 72 h. Egg hatching inhibition percentage was calculated using the following formula:

$$\text{Hatching inhibition \%} = \frac{\text{Number of juveniles hatched in control} - \text{Number of juveniles hatched in treated}}{\text{Number of juveniles hatched in control}} ) \times 100$$

The mortality percentage was calculated from the following formula:

$$\text{Mortality } (\%) = \left( \frac{\text{dead juveniles in treatment} - \text{dead juvenbiles in the negative control}}{(100 - \text{dead juveniles in the negative control})} \right) \times 100$$

### 2.6. Design of the Greenhouse Experiment

In vivo, a pot experiment was conducted to study the effects of n-HAP and mycorrhizal fungi on tomato plants' growth, nutrient uptake, and *M. incognita* infection in the greenhouse at the Plant Pathology Department, Faculty of Agriculture, Assiut University, Assiut, Egypt. Tomato seedlings were planted in sterilized pots (20 cm diameter) filled with a 3 kg mixture of clay: sand (1:2, v:v) and each pot contained three tomato seedlings grown under natural greenhouse conditions and regularly irrigated. The physicochemical parameters of the soil used were pH 7.6, organic matter (%) 0.73, available P (mg/kg) 7.3, total N (mg/kg) 12.3, $Ca^{++}$ (mg/100 g) 1.25, $Mg^{++}$ (mg/100 g) 2.75, and $K^+$ (mg/100 g) 0.16.

The greenhouse experiment was consisted of 7 treatments as follow: (1) healthy plants, without any inoculation (C); (2) plants inoculated with arbuscular mycorrhizal fungi (AMF) (M); (3) plants infected with 2500 J2s (N); and (4) plants treated with HAP nanoparticles (Np); (5) M + N; (6) Np + N; (7) M + NP + N. The mycorrhizal soil inoculum was added during planting and mixed with the soil surface. Plants were treated by 2500 J2s /pot, by pouring the suspension into three holes 2.5 cm deep in the soil around the base of plants. Additionally, 100 mL of HAP nanoparticle solution (200 ppm) was incorporated into the soil surface around plants only once. Each treatment was achieved in four replicates and arranged in a completely randomized design. The control plants received the same weight as sterilized soil.

### 2.7. Parameters' Assessment

### 2.7.1. Plant Analysis

The effect of n-HAP, mycorrhizal fungi, and *M. incognita* on tomato plants' growth parameters, root length (cm), shoot length (cm), plant dry weight (g/plant), and leaf area ($cm^2$) were assessed in 56-day-old plants. The leaf area of tomato plants ($cm^2$) was calculated from leaf images taken with a standard scanner using the ImageJ program (https://imagej.nih.gov/ij/, accessed on 20 December 2021).

Shoots and roots (N, P, Ca) were determined in plant tissues that were oven-dried for 48 h at 70 °C. Then, shoots and roots were powdered and digested in $HNO_3$:$HClO_4$ (2:1 *v/v*) for P and Ca concentration using inductively coupled plasma mass spectrometry (ICP-MS). The nitrogen (N) content was assessed in the digested plant's tissues with $H_2SO_4$ following the Kjeldahl method.

### 2.7.2. *Meloidogyne incognita* Assay

After eight weeks from RKN inoculation, plants were removed, and their roots were gently washed to remove adhering substrate. The J2s was extracted from the soil (250 g), and they were counted in each treatment by decanting and sieving technique [30]. The roots were stained in 0.01% acid fuchsin [31] and examined for the developmental stages (DS, juveniles in the third and fourth stages that succeeded in penetrating the roots), females, galls, and egg masses using the stereomicroscope. To count eggs, three egg masses were randomly selected from each root by shaking with 1% sodium hypochlorite and counted [32]. Root gall index (RGI) was determined according to Taylor and Sasser [33]. The rate of nematode build-up was calculated according to the formula (Pf/Pi), where Pi is the initial nematode population and Pf is the final population. The final population and percentages of reduction in nematodes were calculated by the following formulas:

$$\text{Pf} = (\text{no. Egg/masses per root} \times \text{no. Egg} - \text{masses}) + (\text{Development stages/root} + (\text{Juveniles in soil }) + (\text{Adult females/root})$$

$$\text{Reductuion\%} = \left( \frac{\text{Pf in control} - \text{Pf in treatment}}{\text{Pf incontrol}} \right) \times 100$$

### 2.7.3. Assessment of Mycorrhizal Parameters

To measure the level of mycorrhizal root colonization, tomato plants' roots were uprooted and gently washed using tap water to remove the soil particles. Then, the roots were cut into 1–2-cm segments. Cleaned segments (30 root segments from each replicate) were softened in KOH (10%) for about 30 min, washed in sterilized water, and then acidified in HCl (1%) for 30 min at room temperature. Then, tomato plants' roots were stained by 0.05% trypan blue according to the method of Philips and Hayman [34]. Consequently, the assessment of AMF colonization was performed using the Mycocalc software following the method described by Trouvelot et al. [35] (http://www.dijon.inra.fr/mychintec/Mycocalc-prg/download.html, accessed on 25 December 2021). The mycorrhizal frequency (F%), the intensity of mycorrhizal colonization (M%), relative arbuscules richness (A%), and vesicles (V%), were calculated in the mycorrhizal-tomato plant roots.

### 2.8. Statistical Analyses

All data were subjected to analysis of variance (ANOVA) using the Costat package. Means and significance were calculated according to Duncan's multiple range tests at $p < 0.05$. Correlation and regression studies (equations and trend lines) implemented using IBM SPSS statistics v.21, by regression analysis Pearson product moment correlation type, which stipulates coefficient ('r'), is a measure of the linear association of two independent variables. If the probability that r = 0 ('P(r = 0)') is ≤0.05, r is significantly different from 0 and the variables show some degree of correlation.

## 3. Results

### 3.1. Characterization of Hydroxyapatite Nanoparticles (n-HAP)

The XRD pattern of n-HAP is shown in Figure 1. As can be observed from this figure, n-HAP exhibits several diffraction peaks at 2θ values of 25.80, 28.90, 31.80, 32.80, 34.0, 39.80, 46.70, 49.50, and 53.20 correspond to the (002), (210), (211), (112), (300), (202), (222), (213), and (004) crystal planes of the n-HAP, respectively. These diffraction peaks can be assigned to hexagonal crystal structures with space group P63/m according to the standard JCPDS card no. 01-0721243. All the diffraction peaks observed are characteristics of n-HAP, and peaks corresponding to other calcium phosphate phases or impurities were not detected. The average crystallite size was estimated using the Scherrer formula as follows:

$$D = \frac{0.9\lambda}{\beta \cos \theta}$$

where D is the crystallite size (nm), λ is the wavelength of the X-ray beam (λ = 0.154056 nm for Cu Kα radiation), β is the full width at half maximum for the diffraction peak under consideration, and θ is the corresponding diffraction angle. The average crystallite size of the n-HAP was calculated as 15 nm.

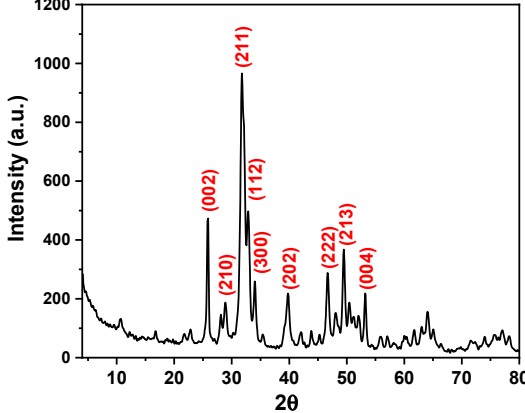

**Figure 1.** X-ray diffraction (XRD) pattern of nano-hydroxyapatite (n-HAP).

The FTIR spectrum of n-HAP is presented in Figure 2. The n-HAP showed vibration bands located at 3570 cm$^{-1}$, 3420 cm$^{-1}$, 1091 cm$^{-1}$, 1031 cm$^{-1}$, 961 cm$^{-1}$, 632 cm$^{-1}$, 601 cm$^{-1}$, 563 cm$^{-1}$, and 473 cm$^{-1}$. The stretching vibration of the O-H due to adsorbed water molecules is observed at 3420 cm$^{-1}$. TEM analysis of n-HAP (Figure 3) revealed that n-HAP was composed of rod-like particles with an average width of 28.5 nm and an average length of 74 nm.

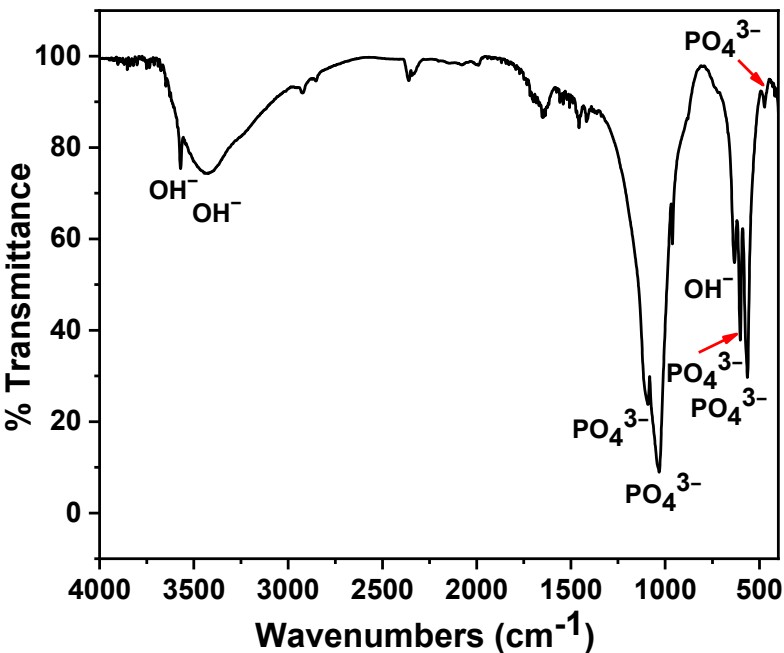

**Figure 2.** Fourier transforms infrared (FTIR) spectrum of nano-hydroxyapatite (n-HAP).

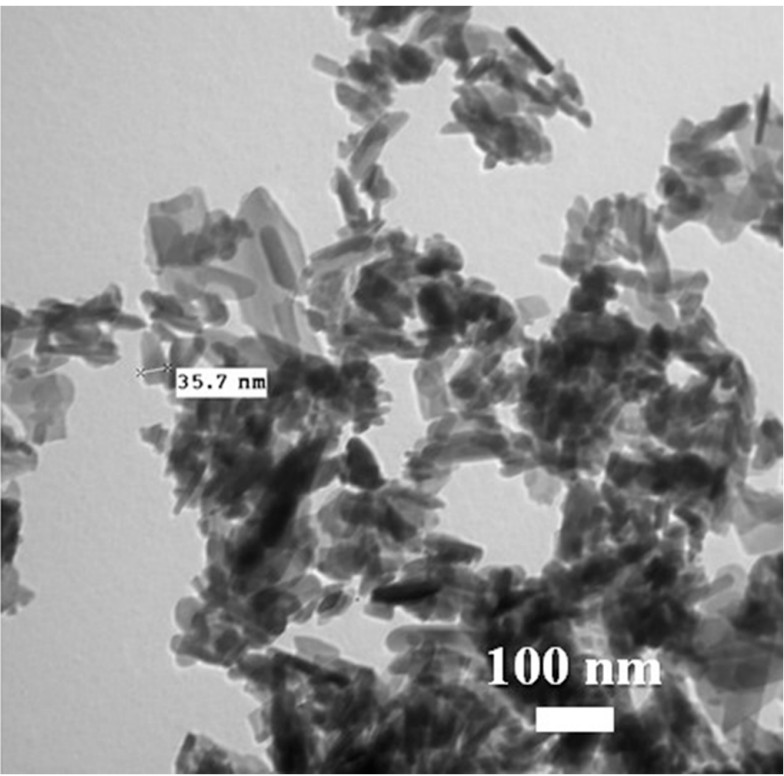

**Figure 3.** Transmission electron microscopy (TEM) image of nano-hydroxyapatite (n-HAP).

### 3.2. Effect of n-HAP on Egg Hatching and Juvenile Mortality of Meloidogyne incognita In Vitro

The effect of n-HAP, at concentrations 50, 150, and 200 ppm, on the egg hatching of *M. incognita* was evaluated under laboratory conditions (Table 1). The data showed that all tested concentrations significantly reduced ($p < 0.05$) the percentage of egg hatching compared to the untreated control. Generally, the effect of n-HAP varied depending on the exposure time and the concentration. The effect on mortality percentage also revealed that all treatments were characterized by a steady relationship between concentration and exposure time. The obtained data exhibited a normal gradient in the difference in the inhibitory effect of egg hatching between the tested concentrations after 24 h of exposure. The highest concentration involved 80.71% of inhibition, while the lowest concentration caused 59.16% inhibition compared to the untreated control (Table 1). When the exposure time was increased to 72 h, the actual and continuous effect of n-HAP was observed, and the inhibition rate of the three tested concentrations of n-HAP (50, 150, and 200 ppm) reached 62.20%, 72.82%, and 83.80%, respectively. On the other hand, direct exposure to n-HAP seemed to have a faster effect on the percentage of mortality of second-stage juveniles than the effect on egg hatching (Table 2). Among them, after 24 h of exposure, the highest concentration of n-HAP (200 ppm) caused 195.67% mortality of J2s, while the lowest concentration (50 ppm) caused 180.16% mortality compared to the untreated control. Mortality was increased with increasing exposure time to 72 h, resulting in the death of all tested individuals.

**Table 1.** Effect of n-HAP on egg hatching of *Meloidogyne incognita* in vitro.

| Treatments | Nematode Hatching | | | | | |
| --- | --- | --- | --- | --- | --- | --- |
| | Egg Hatching after 24 h | Hatching Inhibition% | Egg Hatching after 48 h | Hatching Inhibition% | Egg Hatching after 72 h | Hatching Inhibition% |
| 50 ppm n-HAP | 42.33 ± 5.03 [b] | 59.16 | 46.33 ± 5.03 [b] | 69.71 | 72.33 ± 21.39 [b] | 62.20 |
| 150 ppm n-HAP | 29.00 ± 8.54 [c] | 72.03 | 40.66 ± 5.51 [b] | 73.42 | 52.00 ± 5.57 [bc] | 72.82 |
| 200 ppm n-HAP | 20.00 ± 5.35 [c] | 80.71 | 24.00 ± 6.68 [c] | 84.31 | 31.00 ± 6.53 [c] | 83.80 |
| Distilled water | 103.67 ± 19.34 [a] | - | 153.00 ± 15.12 [a] | - | 191.33 ± 3.68 [a] | - |
| LSD | 24.95 | - | 116.41 | - | 22.52 | - |
| P | 0.0002 | - | 0.0000 | - | 0.0000 | - |

Each value represents a mean of three replicates ±SD. Values in the same column followed by the same letter are not significantly different according to Duncan's multiple-range test ($p < 0.05$). The least significant difference (LSD).

**Table 2.** Effect of n-HAP on second-stage juvenile mortality of *Meloidogyne incognita* in vitro.

| Treatments | After 24 h | | After 48 h | | After 72 h | |
| --- | --- | --- | --- | --- | --- | --- |
| | Dead J2s | * Mortality% | Dead J2s | * Mortality% | Dead J2s | * Mortality% |
| 50 ppm n-HAP | 170.00 ± 3. 04 [b] | 180.16 | 191.67 ± 9.10 [a] | 210.01 | 200.00 ± 0.00 [a] | 238.26 |
| 150 ppm n-HAP | 193.33 ± 9.07 [a] | 193.33 | 199.00 ± 1.73 [a] | 199.00 | 200.00 ± 0.00 [a] | 200.00 |
| 200 ppm n-HAP | 195.67 ± 7.51 [a] | 195.67 | 200.00 ± 0.00 [a] | 200.00 | 200.00 ± 0.00 [a] | 200.00 |
| Distilled water | 12.67 ± 3.06 [c] | - | 16.67 ± 4.73 [b] | - | 27.67 ± 11.50 [b] | - |
| LSD | 36.95 | - | 11.517 | - | 10.829 | - |
| P | 0.0000 | - | 0.0000 | - | 0.0000 | - |

Each value represents a mean of three replicates ±SD. The initial number of juveniles = 200 freshly hatched second stage juveniles. * Increasing mortality over control. Values in the same column followed by the same letter are not significantly different according to Duncan's multiple-range test ($p < 0.05$). The least significant difference (LSD).

### 3.3. In Vivo Effect of n-HAP and Mycorrhizal Fungi on Tomato Plants Infected with M. incognita

#### 3.3.1. Plant Biomass

Table 3 shows the results of growth parameters of tomato plants infected with *M. incognita* and inoculated with mycorrhizal fungi and/or n-HAP. All growth parameters of mycorrhizal inoculated plants were significantly improved compared with the non-mycorrhizal plants. The reduction in plant growth parameters caused by nematode infection was significantly decreased in mycorrhizal plants compared to the nematode-infected plants. The lowest value of root and shoot length was recorded for the infected plants (N), of which the infection decreased the length by 14.51% and 26.30%, respectively, from the control.

Meanwhile, mycorrhizal inoculation increased root and shoot length by 21.79% and 69.50%, respectively, compared to the control. Further, n-HAP treatment increased plant length, and the highest dry weight of the plants was achieved in the dual-treated plants and infected with the nematode (MNpN). Mycorrhizal inoculation increased the plant's dry weight, whereas the nematode infection had no significant effect on the dry weight. Application of HAP nanoparticles significantly increased plants' dry weight by 40.46% compared to the control. There was a significant difference in the plant leaf area among the treatments. Nematode infection significantly decreased the leaf area and recorded the lowest value as 33.89 cm$^2$. Mycorrhizal and/or n-HAP treatments increased the leaf area and reduced the negative effect of nematode infection. The results showed that mycorrhizal inoculation and dual treatments increased the leaf area by 42.46% and 39.30%, respectively, compared to the control.

**Table 3.** Effect of mycorrhizal inoculation and n-HAP on the growth of 56-day-old tomato plant infected with *M. incognita*.

| Treatment | Root Length (cm) | Shoot Length (cm) | Dry Weight (g) | Leaf Area (cm$^2$) |
|---|---|---|---|---|
| C | 20.50 ± 1.29 [a] | 39.35 ± 2.05 [b] | 1.21 ± 0.30 [a] | 36.25 ± 0.93 [a] |
| M | 34.75 ± 2.07 [c] | 47.93 ± 2.39 [c] | 1.78 ± 0.26 [bc] | 51.65 ± 2.71 [c] |
| N | 17.52 ± 1.18 [a] | 29.00 ± 2.29 [a] | 1.05 ± 0.33 [a] | 33.89 ± 1.68 [a] |
| Np | 27.00 ± 1.70 [b] | 46.75 ± 1.89 [c] | 1.70 ± 0.42 [b] | 44.22 ± 1.56 [b] |
| MN | 33.77 ± 1.46 [c] | 37.02 ± 1.67 [b] | 2.29 ± 0.36 [d] | 49.67 ± 1.51 [c] |
| NpN | 21.67 ± 1.48 [a] | 53.50 ± 2.19 [b] | 2.15 ± 0.31 [cd] | 42.76 ± 1.45 [b] |
| MNpN | 37.98 ± 1.83 [c] | 48.10 ± 1.07 [c] | 2.39 ± 0.24 [d] | 50.50 ± 1.05 [c] |

Values followed by the same letter(s) in the same column are not significantly different at $p < 0.05$ using Duncan's multiple range test. Healthy plants without any inoculation (C); plants inoculated with AMF (M); plants infected with *M. incognita* (N); plants treated with HAP nanoparticles (Np); plants infected with *M. incognita* and inoculated with AMF (MN); plants infected with *M. incognita* and treated with n-HAP (NpN); plants infected with *M. incognita* and treated with AMF and n-HAP (MNpN).

### 3.3.2. Effect of Mycorrhizal and n-HAP Treatments on Nutrients' Status in Tomato Plants

The impact of the n-HAP addition and mycorrhizal inoculation on nitrogen, phosphorus, and calcium content in tomato plant's leaf and root was analyzed on plants infected with the nematode (Figure 4). In general, N, P, and Ca levels in tomato plants was reduced by the nematode's infection. The inoculation with mycorrhizae alone or in combination with n-HAP improved roots' and leaves' uptake of the nutrients compared with healthy control (untreated plants), while *M. incognita* infection significantly ($p < 0.05$) decreased the nutrients' uptake to the lowest value among the treatments. Figure 4A showed that mycorrhizal inoculation caused an increase in N concentration in the roots and leaves by 18.00% and 13.00%, respectively, over the healthy control. The concentration of N reached the maximum content when the soil was amended with mycorrhizal treatment or dual treatments. Phosphorus content was significantly increased by the mycorrhizal inoculation and n-HAP (Figure 4B). This improvement was indicated by a 70.90% increase in P content in roots and 54.90% in leaves of MNpN-treated plants compared to the healthy plants. Phosphorus content recorded the highest value in dual-treated plants infected with nematode, while the lowest value was recorded in only nematode-infected plants. There was a 29.19% and 36.50% decrease in P content in nematode-infected plant roots and leaves, respectively, compared to healthy control. The application of n-HAP significantly increased the P content in the roots and leaves of plants by 62.30% and 48.90%, respectively, compared to the untreated control. The highest value of Ca content was detected in the dual-treated plants (Figure 4C), whereas *M. incognita* infection significantly decreased Ca content in roots and leaves by 31.50% and 20.00% compared to the healthy control, respectively. n-HAP significantly increased Ca content in roots and leaves of plants by 43.90% and 41.70%, respectively, compared to the control.

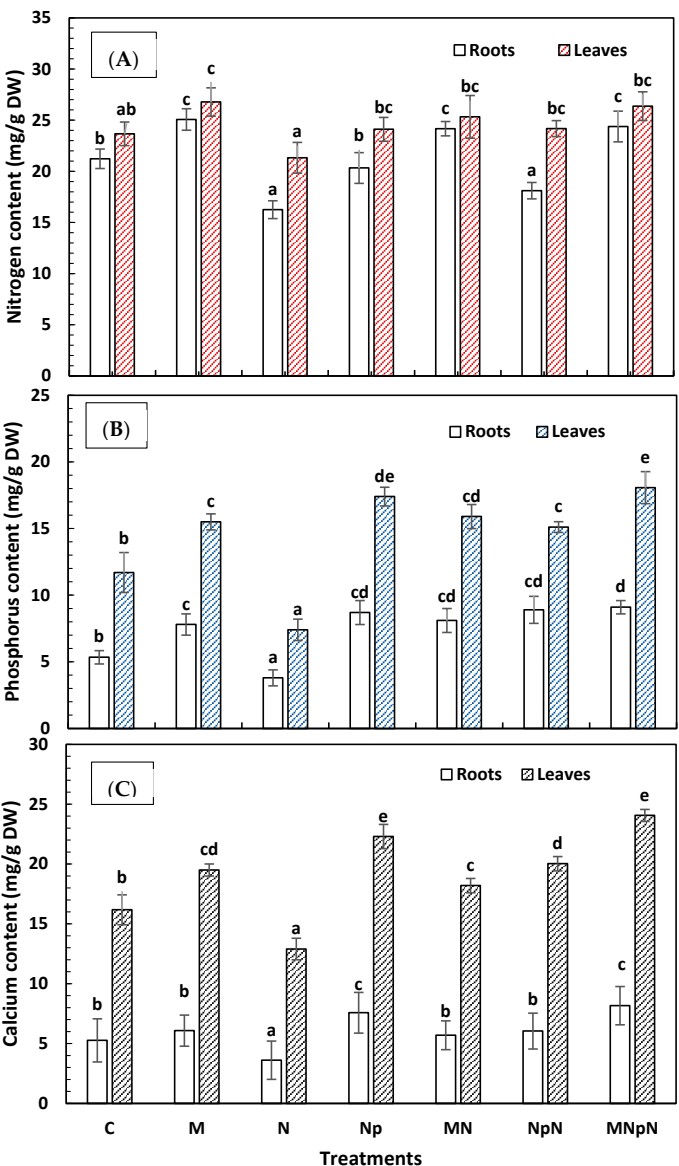

**Figure 4.** Mineral contents (mg/g DW) (**A**) nitrogen, (**B**) phosphorus, and (**C**) calcium in roots and leaves of healthy and *M. incognita*-infected tomato plants inoculated with mycorrhizal fungi and treated with n-HAP. Healthy plants without any inoculation (**C**); plants inoculated with AMF (M); plants infected with *M. incognita* (N); plants treated with HAP nanoparticles (Np); plants infected with *M. incognita* and inoculated with AMF (MN); plants infected with *M. incognita* and treated with n-HAP (NpN); plants infected with *M. incognita* and treated with AMF and n-HAP (MNpN). Vertical bars indicate ± SE (*n* = 4); columns in the same pattern and followed by different letters are significantly different at *p* < 0.05.

### 3.3.3. Effect of Mycorrhizal Fungi and n-HAP on Nematode Parameters

The influence of mycorrhizal fungi and n-HAP, alone or in combination, on *M. incognita* infecting tomato plants, was evaluated under greenhouse conditions. Data in Figure 5 and Table 4 revealed that the nematode criteria (numbers of root galls, the total number of nematodes in different DS that invaded the plants, nematodes in soil, number of egg-mass, number of eggs per egg-mass, final population, rate of nematode reproduction and nematode reduction percentages) were significantly reduced (*p* < 0.05) in treated plants compared with the *M. incognita* infecting tomato plants. The treatments delayed the formation of galls, which eventually resulted in small and soft galls with long distances among the galls, as opposed to the infected control (N). The combination of mycorrhizal

fungi and n-HAP (MNpN) reduced the gall formation (RGI) to 3.33 (93.39%). The number of juveniles in the soil was significantly lower than the developmental stages that succeeded in penetrating the tomato plants treated with n-HAP (Table 4), in which n-HAP resulted in a considerable reduction in the total number of root-knot juveniles in 250 g soil (887.33), followed by mycorrhizal fungi (1013.67) and dual treatments (1228.33). A similar pattern of treatments was shown in the case of egg masses on roots. The shape of egg-laying females was relatively compact in dual treatments, which achieved the best effect in this regard. Regarding nematode build-up and percentage of reduction in the final population, the combined treatments resulted in a significant reduction (2.91%, 93.39%), followed by n-HAP (3.62%, 92.22%) and the mycorrhizal fungi (3.63%, 91.73%), respectively (Table 4). Data in Figure 5a shows a positive and significant strong correlation between treated plants by mycorrhizal fungi (r = 0.973) and positive and significant medium correlation in the plants treated with mycorrhizal fungi and nanoparticles (r = 0.564) (Figure 5c), while results obtained in Figure 5b revealed a negative correlation in the treated plants by nanoparticles only (r = −0.218).

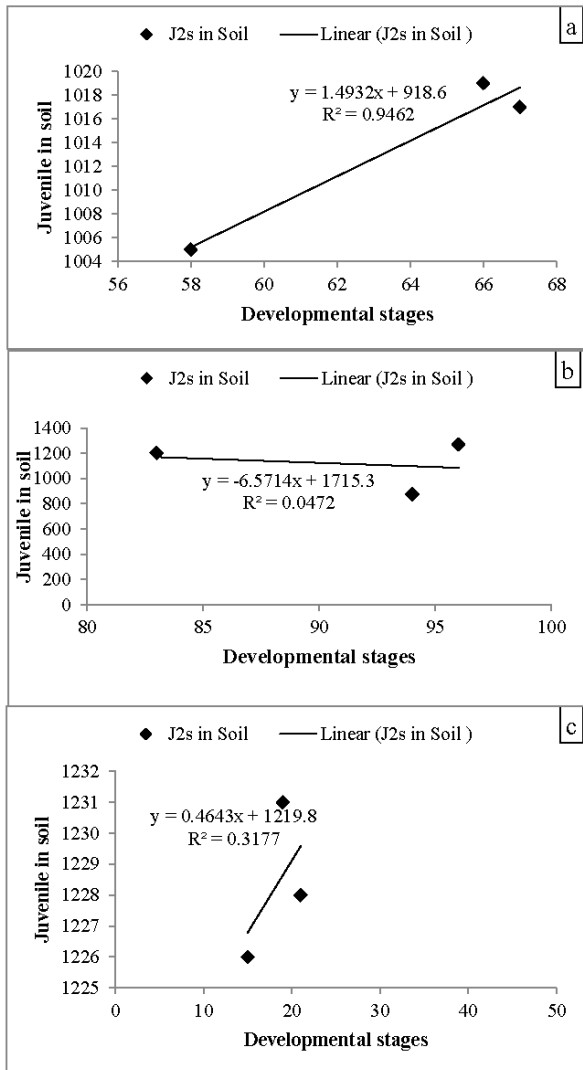

**Figure 5.** Correlation and regression equations juvenile in soil and developmental stages that succeeded in penetrating the roots, were shows (**a**) positive and significant strong correlation in treated by mycorrhizal fungi (r = 0.973), while (**b**), shows a negative correlation in treated by n-HAP (r = −0.218). And (**c**) positive and significant medium correlation in treated by mycorrhizal fungi and n-HAP (r = 0.564).

**Table 4.** Effect of mycorrhizal inoculation and n-HAP on *M. incognita* under greenhouse conditions.

| Treatments | No. of Galls/Root System | No. of Juveniles in Soil (250 g) | Build-Up | RGI ** | Reduction% |
|---|---|---|---|---|---|
| N | 301.67 ± 63.88 [a] | 1406.33 ± 191.06 [a] | 43.93 | 5.00 | - |
| MN | 84.33 ± 4.62 [b] | 1013.67 ± 7.57 [bc] | 3.63 | 4.00 | 91.73 |
| NpN | 62.67 ± 4.73 [c] | 887.33 ± 211.75 [c] | 3.62 | 4.00 | 92.22 |
| MNpN | 27.33 ± 4.04 [d] | 1228.33 ± 2.52 [ab] | 2.91 | 3.33 | 93.39 |

Each value represents the mean of three replicates ±SD. ** Root Gall index (RGI) was scored according to Taylor and Sasser (1978). Values in the same column and followed by the same letter are not statistically different according to Duncan's multiple-range test ($p < 0.05$). Nematode-infected plants (N); plants infected with *M. incognita* and inoculated with AMF (MN); plants infected with *M. incognita* and treated with n-HAP (NpN); plants infected with *M. incognita* and treated with AMF and n-HAP (MNpN).

### 3.3.4. Mycorrhizal Colonization Pattern

Mycorrhizal colonization of tomato plants inoculated with mycorrhizal fungi and n-HAP and infected with *M. incognita* was determined (Figures 6 and 7). The tomato plant roots of all mycorrhizal-inoculated plants were colonized by AMF hyphae. Mycorrhizal hyphae were not detected in the un-inoculated plant roots. Mycorrhizal plant roots were occupied by inter-, intra-, and extra-radical hyphae, coils, arbuscules (Arum-type), and different shapes of vesicles (Figure 6). Also, spores of *Rhizophagus aggregatus* colonized in the plant roots were observed (Figure 6G). There was a significant difference in mycorrhizal colonization percentage among the treatments. Mycorrhizal colonization was affected by the nematode's infection and n-HAP. The percentage of mycorrhizal frequency (F%), which indicated the importance of tomato plant host colonization by AMF, recorded the lowest value in plants infected with *M. incognita* and treated with n-HAP, while the highest F% value was detected in the mycorrhizal tomato plant roots infected with *M. incognita*. The mycorrhizal intensity (M%) values were significantly different among the treatments, which reflected the proportion of tomato plants' roots colonized by AMF. The highest M% value was recorded in MN-plants (mycorrhizal plants infected with nematodes). Arbuscular and vesicle colonization were significantly lower in nematode-infected plants and those treated with mycorrhizal fungi and n-HAP (MNpN) (Figure 7).

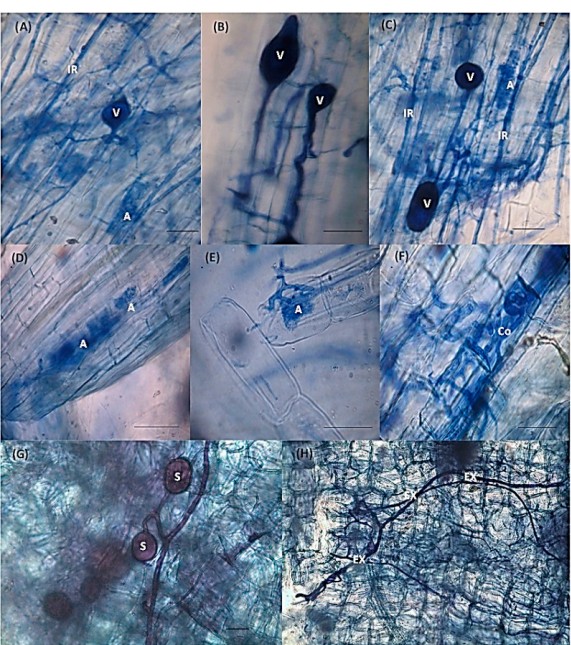

**Figure 6.** Mycorrhizal root colonization patterns of 56-day-old tomato plants inoculated with mycorrhizae and n-HAP and infected with *M. incognita*: (**A–C**) showed vesicles "V", intraradical hyphae "IR", and arbuscules "A"; (**D,E**) showed arum-type of arbuscules; (**F**) showed hyphal coils "Co"; (**G**) showed spores of *Rhizophagus aggregatus* colonizing the host tomato root "S"; and (**H**) showed extraradical hyphae "EX". Bar, 50 μm.

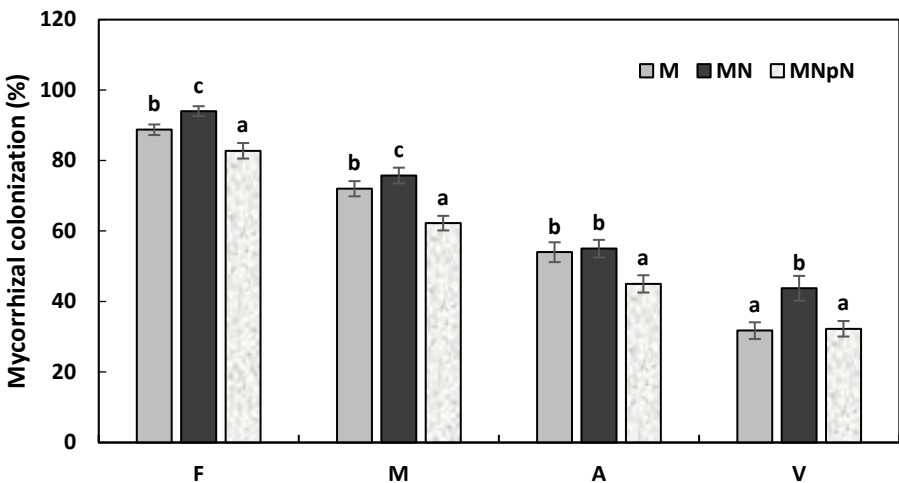

**Figure 7.** Mycorrhizal colonization of 56-day-old tomato plants inoculated with mycorrhizae and n-HAP and infected with *M. incognita*. Mycorrhizal parameters: F-mycorrhizal frequency, M—the intensity of mycorrhizal colonization, A—relative arbuscule richness, and V—vesicles. Within each mycorrhizal parameter, levels that are not connected with the same letter indicate statistically significant difference ($p < 0.05$). Plants inoculated with AMF (M); plants infected with *M. incognita* and inoculated with AMF (MN); plants infected with *M. incognita* and treated with AMF and n-HAP (MNpN).

## 4. Discussion

Plants may be damaged by plant-parasitic nematodes in different ways, either directly by penetration or indirectly through the feeding process via interfering with water and nutrient absorption and transport [36]. Application of the beneficial microorganisms, especially symbionts with plant-parasitic nematodes in soil and hosts or phyto-nanotechnology represents a promising trend in raising plant health to a level that exceeds the limit of damage caused by nematodes' invasion [10,20,37]. In the current study, we reported for the first time the effect of hydroxyapatite nanoparticles and mycorrhizal fungi on *M. incognita* infection. Also, our study revealed a promising observation about the applicability of HAP nanoparticles as nano-fertilizers in plant nutrition, as a source of available P and Ca. The inhibition assessment for nematodes indicated that HAP nanoparticles were highly effective against *M. incognita*. Also, mycorrhizal colonization played an important role in tomato plant growth and plant nutrient contents in case of nematode infection.

The synthesis and characterization of the n-HAP used in this study were already reported [19,38]. In this work, n-HAP were dissolved in sterilized distilled water and then shaken to obtain a stable n-HAP colloidal suspended nanoparticle solution. Meanwhile, the aggregation in the dry state seen in the TEM micrograph is due to the dispersion of the electrostatic interactions of the surface of nanoparticles [19]. The FTIR analysis showed the bands that were located at 1091 cm$^{-1}$, 1031 cm$^{-1}$, 961 cm$^{-1}$, 601 cm$^{-1}$, 563 cm$^{-1}$, and 473 cm$^{-1}$ could be ascribed to orthophosphate ($PO_4^{-3}$) [39], while the bands that appeared at 3570 cm$^{-1}$ and 632 cm$^{-1}$ could be attributed to the O-H stretching and vibrational mode of the structural hydroxyl group [40].

In the present study, application of different concentrations from n-HAP as a nematicide in vitro showed that all applied concentrations of n-HAP decreased egg hatching and significantly induced second-stage juvenile mortality of *M. incognita*. The obtained results from n-HAP demonstrated that egg hatching and mortality of J2s increased progressively as n-HAP concentrations increased. The nematocidal effect of n-HAP may be due to their elemental constituent concentrations (Ca and P) and their low pH, but not due to their toxicity. On the contrary, the inhibitory toxic effect of different metal nanoparticles (silver nanoparticles, silica oxide nanoparticles, silicon nanoparticles) against *M. incognita* has been illustrated in several reports [41,42]. In addition, many reports have demonstrated that phosphonate fertilizer (calcium phosphate) has a negative effect on egg hatching and the second-stage juveniles of *M. incognita* and *M. javanica* [43,44]. In other authors' work,

4 different fertilizers (NPK, humic acid, agro-health, and azotobarvar-1) were tested for their ability to manage nematode populations in vitro, and the results showed that these fertilizers reduced the egg hatching and increased mortality of the second stage juveniles of root-knot nematodes [45].

In the present study, all growth parameters of mycorrhizal plants were significantly improved, while nematode infection had an inhibitory effect on plant growth. The data from our study suggested that mycorrhizal fungal inoculation could alter nutrient uptake in tomato plants. Nitrogen, phosphorus, and calcium contents changed in response to the mycorrhizal inoculation even under nematode infection. Mycorrhizal species (*R. hizophagus irregularis*, *F. mosseae*, or both) can impact the quality of tomato fruits and enhanced the nutrients' content [46]. Furthermore, the extraradical mycorrhizal hyphae can penetrate the narrow pores in the soil and extend to long distances to absorb essential nutrients for the host plants.

The application of hydroxyapatite nanoparticles in this study improved tomato plant growth and nutrient uptake. HAP nanoparticles were experimentally confirmed as nanofertilizers [47] when stabilized with carboxymethylcellulose and evaluated in tomato seed germination and seedling growth [19]. In that work, n-HAP had non-toxic effects on tomatoes; therefore, it can be used as a phosphorus fertilizer and as a carrier of other plant nutrients. HAP nanoparticles can effectively reduce the bioavailability and mobility of lead, thereby controlling the harmful effect of lead pollution on organisms and the environment [48].

The results of nematode infection revealed that the root-knot nematode *M. incognita* population was significantly reduced by the mycorrhizal fungi and/or n-HAP. In the present study, the highest total number of galls per root system, developmental stages in roots, and females was found on uninoculated and nematode-infected tomato plants. This was consistent with earlier reports on mycorrhizal-induced resistance root-knot nematodes in different plant species [49,50]. Recently, biological control has emerged as a cost-effective and environment-friendly method to manage nematode infection and increase crop yield [51]. Direct competition between mycorrhizal fungi and endoparasitic nematodes is not possible, where mycorrhizal fungi reduce nematodes' infection by inducing systemic defense and changing root host exudates [52]. In our study, we used a mixture from mycorrhizal species (*R. aggregatus*, *F. mosseae,* and *G. gigantea*) to increase the specific mode of action and to improve plant nutrient uptake. *F. mosseae* reduced *M. incognita* population by 45% in tomato plants' roots [49], while *R. irregularis* increased tomato growth, nutrients uptake, and *M. incognita* infection (more galls with multiple egg mass) [51]. This growth improvement of infected tomato plants was probably due to their good acclimation to the nematode infection [51]. Diedhiou et al. [53] mentioned that hosts pre-treated with arbuscular mycorrhizal fungi stimulated the health of the host and reduced infestation of root-knot nematodes. The effect of the tested mycorrhizal fungi may be due to their colonization of the epidermal cells as well as the inner cells of the cortex [54,55]. Also, early colonization of plant roots by hostile fungi can protect the plant from nematode invasion by parasitism, competition, or physical confinement. In addition to the mycorrhizal fungi that can increase nutrients and water uptake by plants, they also incite induced resistance to nematodes and compete on food resources and available space with nematodes [56].

The current study showed that mycorrhizal spores can establish a symbiotic association with tomato host plants under different treatments. This confirmation could be demonstrated by the determination of different mycorrhizal parameters. *F. mosseae* colonized tomato plant roots starting two weeks from planting, and it increased in intensity over time, reaching 91% (F%) and 18% (M%) after 28 days [57]. It was found that, mycorrhizal colonization positively impacted by RKN infection as the highest mycorrhizal colonization parameters was recorded in tomato plants infected with *M. incognita*, while the lowest value of mycorrhizal colonization was recorded in the plants treated with n-HAP. This result was consistent with the hypothesis that fertilizers, especially N and P, can reduce the mycorrhizal colonization [58]. For instance, in wheat plants in which $Ca(H_2PO_4)$ or

DAP was used as P fertilization the mycorrhizal colonization was strongly decreased [59]. Mycorrhizal fungi can overcome the impact of different soil-borne pathogens and plant-parasitic nematodes. Arbuscular colonization in the nematode-infected plants was higher as compared with MNpN plants. The mature symbiosis is imported toward the action of mycorrhizae-induced internal plant resistance against nematode infection [49].

## 5. Conclusions

The current study showed the indirect positive effect of mycorrhizal fungi and HAP nanoparticles on tomato plants infected with root-knot nematode, which inhibited the nematodes' multiplication and galls formation. A synergy was also observed between hydroxyapatite nanoparticles and mycorrhizal fungi as pre-inoculated agents on inoculated plants infected with nematodes. The impact of mycorrhizal fungi and n-HAP, alone or in combination, on *M. incognita*-infected plants revealed a significant increase in plant growth parameters and nutrient content, which may illustrate the enhancement of nutrient use efficiency and, consequently, stimulate systemic plant resistance. The dual inoculation reduced the nematode criteria (numbers of root galls, nematodes in soil, number of egg mass, and rate of nematode reproduction) compared with the control. Also, the negative effect of hydroxyapatite nanoparticles on nematodes may be due to changes in the surrounding environmental properties, which created an inappropriate medium that led to increased nematode mortality. Interestingly, the present work is the first report to assess the impact of hydroxyapatite nanoparticles on nematode infection. The nature of hydroxyapatite nanoparticles as a calcium phosphate compound has increased their feasibility as novel biofertilizer in agricultural technology due to their nano properties, ease of absorption and deposition, and stability under various conditions.

**Author Contributions:** E.A.H., N.A.N., A.M.E.-S. and M.A.E.-A.: experimental design, methodology, software, formal analysis, data curation, writing—original draft preparation, writing—review and editing; M.H. and Y.S.M.: supervision, formal analysis, investigation, data curation, writing—review and editing; S.A.: revised the manuscript, visualization, project administration, funding acquisition. All authors have read and agreed to the published version of the manuscript.

**Funding:** This research was funded by the Deputyship for Research and Innovation, Ministry of Education, Saudi Arabia, under the project number IFP-KKU-2020/2.

**Institutional Review Board Statement:** Not applicable.

**Informed Consent Statement:** Not applicable.

**Data Availability Statement:** Not applicable.

**Acknowledgments:** The authors extend their appreciation to the Deputyship for Research and Innovation, Ministry of Education, Saudi Arabia, for funding this research work through the project number IFP-KKU-2020/2.

**Conflicts of Interest:** The authors declare no conflict of interest.

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
