# Peer review of "Current Utility of Arbuscular Mycorrhizal Fungi and Hydroxyapatite Nanoparticles in Suppression of Tomato Root-Knot Nematode"

_agronomy, doi:10.3390/agronomy12030671_

Round 1
Reviewer 1 Report
The manuscript by Alamri and colleagues reports on the combined use of arbuscular mycorrhizal fungi and hydroxyapatite NPs, and their effect on tomato plant development as well as on nematode.
The study is of clear interest in the perspective of the introduction of more sustainable agricultural practices based on biofertilizers.
Minor points:
Line 131: The author should explain (in the introduction) why they choose to use a mixture of different mycorrhizal fungi instead of applying a single one.
Line 162: The author should determine the chemical composition (N, P, K, ....) of clay and sand used in this study.
Line 169: How many times the HAP NPs were applied to the soil?
Line 207: Please, describe the method used for root staining and how you calculated F%, M%, A%, and V%.
Line 393: In figure 4C, please use the term content instead of conotent.
Line 500: it would be nice if Figure 6 shows not only the roots of M but also those of M+N, M+NPs, and M+N+NPs.
Author Response
Minor points:
Line 131: The author should explain (in the introduction) why they choose to use a mixture of different mycorrhizal fungi instead of applying a single one.
Response:
Thank you very much for the comment, we added the following paragraph to the introduction section to answer about the raised question:
“Native mycorrhizal fungi strains are used as biofertilizers and bioprotective agents to increase plant yields and protect plant from pathogens, as an environmentally friendly agent. The combined co-inoculation of different mycorrhizal species belonging to different families were most efficient than mono-inoculation in mineral nutrition, limited heavy metal transport to shoot and improved the tolerance of plant against biotic and abiotic stress (Yang et al. 2017; Crossay et al. 2019)“
Line 162: The author should determine the chemical composition (N, P, K, ....) of clay and sand used in this study.
Response:
The chemical composition was added in materials and methods as following “The physicochemical parameters of the soil used were pH 7.6, organic matter (%) 0.73, available P (mg/kg) 7.3, total N (mg/kg)12.3, Ca++ (mg/100 g)1.25, Mg++ (mg/100 g)2.75, and K+(mg/100 g) 0.16.”
Line 169: How many times the HAP NPs were applied to the soil?
Response:
HAP Nps were applied one time in the soil. This was mentioned in the appropriate place.
Line 207: Please, describe the method used for root staining and how you calculated F%, M%, A%, and V%.
Response:
The method of root staining was inserted in the text “The cleaned segments (30 root segment from each replicate) were softened in KOH (10%) for about 30 min, washed in sterilized water and then acidified in HCl (1%) for 30 min at room temperature. Then tomato roots were stained by 0.05% trypan blue according to the method of Philips and Hayman [34].
As well, the calculation method of AMF colonization was inserted “the assessment of AMF colonization was performed using the Mycocalc software following the method described by Trouvelot et al. [35] (http://www.dijon.inra.fr/mychintec/Mycocalc-prg/download.html). The mycorrhizal frequency (F %), the intensity of mycorrhizal colonization (M %), relative arbuscules richness (A %) vesicles (V %), and in the mycorrhizal tomato roots were calculated.”
Line 393: In figure 4C, please use the term content instead of conotent.
Response: ok, it was corrected
Line 500: it would be nice if Figure 6 shows not only the roots of M but also those of M+N, M+NPs, and M+N+NPs.
Response:
Thanks for the comment, this figure showed the different patterns of mycorrhizal structures recorded from different treatments (Plants inoculated with AMF (M); plants infected with M. incognita and inoculated with AMF (MN); plants infected with M. incognita and treated with AMF and n-HAP (MNpN)).
Figure 6 legend was modified as “Figure 6: Mycorrhizal root colonization patterns of 56-old tomato plants inoculated with mycorrhizae and n-HAP and infected with M. incognita: [A-C] Vesicles (V), intraradical hyphae (IR), and arbuscules (A); [D&E] Arum-type of arbuscules (A); [F] Hyphal coils (Co); [G] Spores of Rhizophagus aggregatus colonized in the host tomato root (S); and [H] Extraradical hyphae (EX). Bar, 50 µm.”
Reviewer 2 Report
I do find this work interesting and valuable. The manuscript is well written. I did not find any serious mistakes. Some suggestions below:
I did not find the exact results of RGI, rate of nematode build-up, reduction %..... (nematode parameters), which are mentioned in material and methods – lines 195 – 202. They are described only in the text. Could you show them also in the form of Table or Figure?
Line 419 – here you refered to Figure 5a and mentioned developmental stages around the plants treated with n-HAP, while Figure 5a shows “…in treated by mycorrhizal fungi..”.
Moreover could you explain in more detailed way how exactly these correlations and regression were calculated (I mean what was taken into account) and how it is connected with the parameters mentioned in 2.7.2. Meloidogyne incognita assay
Lines 569 – 570 – again, where are these results?
Author Response
I did not find the exact results of RGI, rate of nematode build-up, reduction %..... (nematode parameters), which are mentioned in material and methods – lines 195 – 202. They are described only in the text. Could you show them also in the form of Table or Figure?
Response:
Thank you very much for the comment, a table of nematode criteria was added (Table 4)
Line 419 – here you refered to Figure 5a and mentioned developmental stages around the plants treated with n-HAP, while Figure 5a shows “…in treated by mycorrhizal fungi..”.
Response:
That data was amended in Table 4.
Moreover could you explain in more detailed way how exactly these correlations and regression were calculated (I mean what was taken into account) and how it is connected with the parameters mentioned in 2.7.2. Meloidogyne incognita assay
Response:
The following sentence was added to the figure 5,“Correlations and regression studies; equations and trend lines implemented by the Pearson product moment correlation type which, stipulates coefficient ('r') is a measure of the linear association of two independent variables. If the probability that r=0 ('P(r=0)') is ≤ 0.05, r is significantly different from 0 and the variables show some degree of correlation”.
- Where this was applied to the relationship between the numbers of nematodes penetrates the plant and the number of individuals that stilled in the soil as evidence of the effect of tested treatments on the rate of nematode penetration into the plants.
Lines 569 – 570 – again, where are these results?
Response:
A table of nematode criteria has been added contained the results (Table 4)